# Impact of the COVID-19 Pandemic on Cancer Diagnosis in Madrid (Spain) Based on the RTMAD Tumor Registry (2019–2021)

**DOI:** 10.3390/cancers15061753

**Published:** 2023-03-14

**Authors:** Gregorio Garrido-Cantero, Federico Longo, Javier Hernández-González, Ángel Pueyo, Tomás Fernández-Aparicio, Juan F. Dorado, Javier C. Angulo

**Affiliations:** 1Oficina Regional de Coordinación Oncológica, Consejería de Sanidad, Comunidad de Madrid, Paseo de la Castellana 280, 28046 Madrid, Spain; 2Centro de Investigación Biomédica en Red de Cáncer (CIBERONC), Instituto de Salud Carlos III, Monforte de Lemos 3-5, 28029 Madrid, Spain; 3Instituto Ramón y Cajal de Investigación Sanitaria (IRYCIS), Hospital Universitario Ramón y Cajal, M-607, 28034 Madrid, Spain; 4Departamento de Medicina, Facultad de Medicina, Universidad de Alcalá, Pl. de San Diego, s/n, 28801 Alcalá de Henares, Spain; 5Departamento Clínico, Facultad de Ciencias de la Salud, Universidad Alfonso X el Sabio, Avda. de la Universidad, 1 Villanueva de la Cañada, 28691 Madrid, Spain; 6Fundación para la Investigación e Innovación Biomédica (FIIB) de los Hospitales Universitarios Infanta Leonor y Sureste, 28003 Madrid, Spain; 7Cátedra de I+D+i Biomédica, Universidad Católica San Antonio de Murcia, Guadalupe de Maciascoque, 30107 Murcia, Spain; 8Servicio de Urología, Hospital General Universitario Morales Meseguer, Av. Marqués de los Vélez s/n, 30008 Murcia, Spain; 9PeRTICA Análisis Estadísticos, Av. Leonardo Da Vinci, 8, Getafe, 28906 Madrid, Spain; 10Departamento Clínico, Facultad de Ciencias Médicas, Universidad Europea, 28005 Madrid, Spain; 11Servicio de Urología, Hospital Universitario de Getafe, 28905 Madrid, Spain

**Keywords:** cancer diagnosis, coronavirus disease 2019 (COVID-19), undetected cancer, stage shift

## Abstract

**Simple Summary:**

The COVID-19 pandemic has caused a significant disruption to cancer diagnosis, treatment and prevention. According to standardized incidence ratios, the large volume of undetected cancer cases related to the COVID-19 pandemic in Madrid has not returned yet to the reference levels. Breast and lung cancer screening programs have recovered completely, but other very prevalent malignancies including colorectal, prostate or bladder cancer still remain severely impacted. These figures may have serious consequences in the future regarding both cancer control and survival. In order to mitigate the long-term consequences of the COVID-19 pandemic strategies to recover the backlog of diagnoses need be considered.

**Abstract:**

The coronavirus disease 2019 (COVID-19) pandemic has caused a significant disruption to cancer diagnosis, treatment and prevention worldwide that could have serious consequences in the near future. We intend to evaluate the weight of this backlog on a community-wide scale in Madrid during the period 2020–2021, and whether a stage shift towards the advanced stage has occurred. Cancer diagnoses in the Madrid tumor registry (RTMAD) from 2019–2021 were evaluated. Absolute and percentage differences in annual volume and observed-to-expected (O/E) volume ratios were calculated. Standardized incidence ratios (SIRs) and 95% confidence intervals (CIs) were calculated using the O/E ratio. The SIR for 2020–2021 compared to 2019 was 94.5% (95% CI 93.8–95.3), with unequal gender-specific cancer diagnosis recovery (88.5% for males and 102.1% for females). Most cancer types were underdiagnosed in 2020. The tendency worsened in 2021 for colorectal and prostate cancers (87.8%), but lung cancer recovered (102.1%) and breast cancer was over-diagnosed (114.4%) compared with reference pre-COVID-19 data. These changes have modified the ranking of the most frequent malignancies diagnosed in Madrid. Breast cancer has overtaken colorectal and prostate cancers, displaced to second and third position, respectively. Not only was colorectal cancer diagnosis affected more as a consequence of the COVID-19 pandemic but diagnosis of this malignancy at the advance stage also increased by 3.6% in 2020 and 4.2% in 2021 compared to the reference period of 2019. In summary, there is a large volume of undetected cancer in Madrid caused by the reduced access to care secondary to the COVID-19 pandemic, especially regarding colorectal and prostate cancer. Strategies are needed to recover the backlog of diagnoses and effectively treat these cases in the future and solve the negative impact that will be caused by the diagnostic delay. Analyzing the impact of new diagnoses suffered by each different malignancy and their recovery will help to understand how the future allocation of resources should look.

## 1. Introduction

Cancer is a leading cause of death in every country and one of the most serious barriers to increased life expectancy. The progressive decline in mortality due to stroke and coronary heart disease has led to cancer becoming a prominent cause of death [1]. The coronavirus disease 2019 (COVID-19) pandemic started in 2020 and has since caused millions of excess deaths around the world, in patients with and without cancer, often challenging the attribution of the cause of death [2,3,4]. The pandemic has affected not only mortality rates but also the diagnosis and treatment of cancer. The leading cause for this was reduced access to healthcare due to clinic closures and a fear of COVID-19 exposure, resulting in delays in diagnosis and treatment. These delays may have led to a short-term drop in cancer incidence, which could be followed by an upsurge in advanced-stage disease and, ultimately, increased mortality [5].

Quantifying these and other secondary consequences of the pandemic at the population level will take several years because of the lag in the dissemination of population-based surveillance data on cancer incidence and mortality. However, registry-based cohort studies can facilitate the evaluation of these items as they are an excellent opportunity to monitor cancer figures in a relatively stable population in real time.

The first COVID-19 pandemic wave severely affected Spain during the Spring of 2020 and a state of emergency was declared with a severe nationwide lockdown exerted from March to June 2020. The sanitary system was shocked and, despite telemedicine being expanded and cancer treatment protocols being optimized, hospital-based studies have uniformly confirmed a decrease in cancer diagnosis of 21–37% in the first wave [6,7,8,9,10], which leveled off at 17% in the first pandemic year [11] and 12% after two years [12]. Channeling health resources to COVID-19 patient care required cancelling or postponing many non-COVID-19-related care activities including cancer screening programs, imaging services, interventions and histopathologic evaluations. The gradual reversal of restrictions after lockdown was followed by recurrent COVID-19 outbreaks. New viral variants emerged, and a patient fear of COVID-19 exposure led to an enormous backlog of consultations and procedures that delayed or completely missed the identification of new cancers. The consequences have not been totally evaluated in the mid- and long-term.

The COVID-19 pandemic has had an enormous impact on incident cancer detection worldwide [13,14]. Our objective is to evaluate the weight of this backlog on a community-wide scale during the first two years of the pandemic for the different neoplasia in Madrid. We also intend to analyze the impact on the stage shift towards advanced malignancies likely caused by the delays in new cancer diagnoses. These facts will determine how the future allocation of resources for cancer healthcare recovery needs is provided.

## 2. Patients and Methods

We examined new detections of cancer reported by the 29 hospitals that contribute to the cancer registry for the autonomous community of Madrid (Madrid tumor registry, RTMAD) from 1 January 2019 to 31 December 2021. The population covered by these health institutions was 6.7 million as of 1 January 2022. There are no disparities in access to healthcare in Madrid due to free, universal healthcare programs. RTMAD has collected the secondary data from all of the public hospitals in Madrid since 2012 in a standardized format from all participating centers including quality check (JRC-ENCR Quality Check Software 2.0 version, European Network of Cancer Registries), coherence between incidence and date of birth, cross-validating topography, morphology, grade and laterality and implementing standard multiple primary definitions. All patients with a new cancer declaration were collected according to the registry standards using the International Classification of Disease-O (ICD-O-3.1) codes. New cancer declarations were considered to be a proxy for cancer incidence. The standards were the same for all of the participating centers, enabling a comparison before and after the pandemic era.

The standardized incidence ratio (SIR) was used to determine whether the rate of neoplasms observed in the two years of the COVID-19 pandemic (2020 and 2021) was lower or higher than expected, taking into account the age distribution of the community of Madrid in the periods studied. In fact, the SIR was obtained by dividing the number of observed cases of cancer in the period of 2020–2021 in RTMAD by the number of cases that would have occurred in our community in that same period, applying to the population of this period the specific rates obtained from RTMAD for the year 2019.

The SIR is defined as the number of observed neoplasms divided by the number of expected neoplasms, where ∑P_i_ is the number of expected neoplasms; P_i_ is the individual probability of cancer diagnosis, defined as [exp (logit P_i_)]/[1 + exp (logit P_i_)]; and logit P_i_ is the logistic regression equation α + (β_1_ × *X*_1_) + (β_2_ × *X*_2_) … + (βp × *X*p). The 95% confidence interval for the SIR is defined as [(the number of observed neoplasms ± 1.96) × σ]/∑P_i_, where σ^2^ = ∑P_i_ × (1 − P_i_).

This study used 2019 as a reference period and 2020–2021 as the COVID-19 period. Declarations were grouped by weeks. The counts were averaged every week to the total number of years under study. The interannual variation rate per month that resulted from dividing the number of cases diagnosed per month by the number of cases registered for the same month in the previous year was calculated. If the number of cases registered was lower than that of the previous year, the rate was lower than 100. Conversely, if it was higher, the variation rate was over 100.

To estimate the influence that each COVID-19 wave suffered in Madrid had on cancer registration, the number of new cases of cancers registered in RTMAD during the COVID-19 period was graphically represented in relation to the number of hospital admissions due to COVID-19 in the same contributing institutions.

The annual differences and differences between reference and COVID-19 periods were calculated as a raw number and as a percentage of the baseline annual volume. Annual deficits were calculated for each neoplasia. This was repeated to evaluate the differences in various characteristics such as gender (male or female), age group (<40, 40–70 or >70), type of cancer (most prevalent cancers) and stage (localized, regional or distant).

Observed-to-expected (O/E) ratios were calculated for the annual number of diagnoses performed. Expected incident cancers (E) were estimated by applying 2019-cancer-incidence-specific rates by sex and 5-year age groups to the updated 2020 and 2021 Madrid population pyramids for each tumor site. Incident cancers during the COVID-19 period were observed cancers (O). A negative difference between O and E cancers was used as an estimate of the undetected cancers during the COVID-19 pandemic. Standardized incidence ratios (SIRs) with their 95% CIs were calculated using the O/E ratio, with SIR values below and over 1 indicating an underdiagnosis and overdiagnosis of cancer, respectively. The SIR minus 1 was the percentage variation in the underdiagnosis or overdiagnosis of cancer.

The registration of in situ carcinoma of the uterine cervix was initiated in 2020. It was included for the total count and for the evolution of interannual variation but was excluded for the evaluation of the SIR in cancer diagnosis. The only factor that may have had an influence on the change in the population structure during the years evaluated is precisely the mortality caused by COVID-19 in 2020 and, to a lesser extent, in 2021. No other confusion factor should be expected.

Statistical analysis was performed using RStudio v.1.41717 software.

The study was IRB-approved. Due to its retrospective design and the use of pseudonymized data, consent to participate was not required. No human tissues were involved in this study.

## 3. Results

### 3.1. RTMAD Cancer Registry 2019–2021

Table 1 shows the number of cancer cases diagnosed in RTMD during the years 2019 to 2021, and also gender and age distribution. A relative reduction in the total number of cancers diagnosed was evidenced from 2019 to 2020, and an increase followed in 2021. Differences in statistical significance were observed in the gender distribution, taking the population of the study as a whole, with the difference in males diagnosed with cancer tending to equalize to that of females. Age distribution also showed statistically significant differences, with a progressive increment in age (mean ± SD) during the years 2019–2021.

### 3.2. Annual Evolution and Interannual Monthly Difference

Table 2 shows the annual evolution and the interannual monthly difference. Monthly variation was over 100 during the first two months of 2020 but drastically descended when the COVID-19 pandemic was declared in Spain on March 2020 and did not increase until March 2021, with a modest recovery during the months of June 2020 and December 2020. Noticeably, these two months were coincident with the values after the first and second COVID-19 pandemic waves.

Figure 1 confirms that the influence on cancer registration in Madrid was mirrored by each wave suffered and the drop-outs in cancer diagnoses were caused by peaks in hospital admissions due to COVID-19. All of the institutions participating in RTMAD admitted COVID-19 patients during the pandemic and at the same time continued providing specialized healthcare for the general population, as allowed for by the interference caused by the lockdown and the restrictions caused by the COVID-19 spread.

### 3.3. Standardized Incidence Ratio (SIR) According to Gender and Age

The adjusted incidence rate was 482 for 2019 (625 males and 380 females), 423 for 2020 (522 male, 354 female) and 485 for 2021 (585 male, 419 female). The SIR for the COVID-19 period compared to the reference was 94.5 (95% CI 93.8–95.3). With respect to gender, it was 88.5 (95% CI 87.5–89.5) for males and 102.1 (95% CI 100.9–103.3), thus confirming unequal gender-specific cancer recovery. Figure 2 presents the SIR for 2020–2021 compared to 2019 per trimester, regarding both the total and according to gender. 

Noticeably, the recovery of the diagnostic backlog according to the standardized incidence was lower for males than females throughout the COVID-19 period. Males did not recover up to the 2019 level until the fourth trimester of 2021, while women recovered earlier in the fourth trimester of 2020. In the total population, with both genders taken together, the SIR equivalent to 2019 was reached in the second trimester of 2021. Compensating for overdiagnoses did not occur in males during the COVID-19 period analyzed, but for females it occurred in the second and fourth trimesters of 2021, coincident with the values in hospital admission of the fifth and sixth COVID-19 pandemic waves in Madrid, evidenced in Figure 1.

The evolution of the SIR change was different according to age, both for males and females. Figure 3 shows the recovery of diagnosis for the different age groups in 2020 and 2021, taking males and females separately. The SIR equivalent to 2019 was confirmed only for younger males (<40 years) in 2021, while other male age groups remained underdiagnosed throughout the COVID-19 period evaluated. On the other hand, female recovery of the backlog with overdiagnosis also took place in 2021 both for the middle (40–70 years) and old (>70 years) age groups, but not in younger patients (<40 years).

### 3.4. Standardized Incidence Ratio (SIR) for Different Neoplasia

Regarding the type of cancer, i.e., the organ involved, during 2020, most of the cancer sites were underdiagnosed. Table 3 confirms that the SIR was under 1 in breast cancer (C50), colorectal cancer (C18–C20), prostate cancer (C61), lung cancer (C34), kidney and urinary tract malignancies (C64, C66–C68), hematologic malignancies (C42, C77), skin cancer (C44), gastric cancer (C16), oral–pharyngeal cancer (C00–C14), thyroid cancer (C73), brain malignancies (C70–C72), ovarium cancer (C56), bone cancer (C40–C41), adrenal and endocrine malignancies (C74, C75) and penile cancer (C60–C63), in descending order of frequency. The SIR remained unchanged in pancreatic cancer (C25), endometrial cancer (C54), laryngeal cancer (C32), esophageal cancer (C15), testis cancer (C62) and several other less frequent malignancies. Overdiagnosis was evidenced during 2020 only in malignancies of an unknown site (C80) and in vulvar cancer (C51–C52). During 2021, the pattern of diagnosis remained affected and continued to reduce, even compared to 2019 data, in some of the most frequent malignancies such as colorectal cancer (C18–C20), prostate cancer (C61) and kidney and urinary tract malignancies (C64, C66–C68). Other very frequent malignancies such as lung cancer (C34) or hematologic neoplasia (C42, C77) recovered their SIR equivalent to that of 2019, which was also the case for pancreatic cancer (C25), stomach cancer (C16), endometrial cancer (C54) and several other less frequent neoplasia. Notably, the malignancies with overdiagnosis, as compared to the pre-COVID-19 data in 2019, were breast cancer (C50), skin cancer (C44), retroperitoneum malignancies (C48–C49), cervix cancer (C53), vulvar cancer (C51–C52) and renal pelvis cancer (C65) (Table 3).

### 3.5. Evolution Trends for the Most Frequent Malignancies

Among the most frequently registered neoplasia, only breast cancer (females) and lung cancer (both sexes) had increased diagnosis rates in 2021 compared to 2019 data. Other malignancies within the top five (colorectal, prostate and urinary bladder) are still far from reaching an SIR of 1 (Figure 4). These changes imply a modification in the ranking of the most frequent malignancies registered in RTMAD during the COVID-19 pandemic period, as evidenced in Table 4.

The prevalence of breast cancer has overtaken colorectal and prostate cancer, which have been displaced to second and third position, respectively. Lung and urinary bladder cancer retain their fourth and fifth place in the global ranking. For males, a reduction in the diagnosis of colorectal cancer during the COVID-19 pandemic period and an increase in the diagnosis of lung malignancy are evidenced. However, despite the underdiagnosis of prostate cancer, it remains in first position. For females, the figures remain more stable with a very interesting increase in the numbers of breast and lung cancer.

The tumor stage at presentation was evaluated for the most frequent malignancies, with absent data per year varying from 0.3 to 3.5 percent of the registries. Discrete changes were observed in the proportion of locally advanced or metastatic cases at presentation during the COVID-19 pandemic period. Advanced colorectal cancer increased by 3.6% in 2020 and 4.2% in 2021 over the reference 2019 period. Figure 5 shows a statistically significant increment in colorectal cancer (*p* = 0.0001, Cochran–Armitage trend test). The discreet increment in advanced forms of lung and prostate cancers did not reach statistical significance (*p* = 0.08 and *p* = 0.31, respectively). On the contrary, a statistically significant decrease in advanced breast malignancy was confirmed (*p* = 0.002).

## 4. Discussion

The global shutdown declared in March 2020 forced many medical and surgical specialties to reorganize their routines, prioritizing diagnostic tests, surgeries and other activities and deferring non-urgent pathologies in the wake of the limited operating rooms and hospital beds occupied by, or reserved for, the avalanche of patients with severe respiratory symptoms needing urgent medical care. The mutational capacity of the virus gave rise to different variants and several repeated pandemic waves. Social distancing was a useful strategy for curbing the spread of COVID-19, especially before the availability of vaccines. However, isolation from healthcare settings due to the fear of infection brought about a decline in the ordinary consultation of alarming signs or symptoms and in the follow-up of patients at risk or with a previous history of cancer [15]. Screening programs were also affected, delayed and sometimes cancelled. Disparities have been observed in the recovery of cancer screening tests after the first peak of the COVID-19 pandemic, depending on the redistribution of hospital resources and the availability of home-based alternatives for the diagnostic elements used [16,17,18,19]. Additionally, the clinical management of everyday practice concerning cancer diagnosis and treatment has been somewhat modified according to the evolution of the pandemic [20].

Obviously, the isolation from cancer prevention and early detection activities has more seriously affected the older population, considered a fundamental target for the prevention of COVID-19 contagiousness, at least in the initial phase of the pandemic. This has led to a more severe extent of cancer underdiagnosis for older individuals, already confirmed in another study performed in the Spanish territory [9]. Additionally, COVID-19 mortality has been more severe in older individuals, which could also have consequences in the diagnosis of cancers that are more frequent in old-aged patients, e.g., prostate cancer.

We carefully evaluated the heterogeneous behavior for age and gender groups in cancer registration during the COVID-19 period and detected not only age but also gender differences in standardized incidence rate recovery. Females in the middle and higher age groups completely recovered their backlog in 2021, compared to pre-COVID-19 data, but males did not. The evolution of cancer diagnosis by gender has already been presented in Figure 2. Cancer diagnosis in females progressively recovered from the fourth term of 2020, while males have not recovered yet.

Paradoxically, Figure 2 and Figure 3 reveal that older males recovered better than low- and middle-age males, and reached the reference pre-COVID-19 standards but without any evidence of compensating overdiagnoses so far. This could be caused by the difference in the oncologic healthcare access but could also be caused by a different recovery in the inherent diagnostic pathway of each type of malignancy. As far as we know, an organized strategy to quantify and solve the deficit in undiagnosed cancers has not been developed. This study could provide the first step in this regard.

Older patients seem to have adjusted their sanitary activities, at least during lockdowns, more seriously than younger age groups. This could have affected the diagnosis of prostate cancer to a higher extent than that of other malignancies. Additionally, neoplasia with diagnoses more dependent on complex procedures (e.g., colonoscopy, cystoscopy and ureteroscopy) may have caused a higher diagnostic delay. To make matters worse, the different screening programs for colorectal and breast cancers have suffered differently from the consequences of the COVID-19 pandemic, not only during the initial lockdown but also later [21]. Differences in the reduction in cancer screening for each particular neoplasia have been recently confirmed [22], and the accumulated backlog of procedures and resultant undiagnosed cancers due to the pandemic likely show a similar trend to those seen for diagnostic and screening procedures [23,24].

On the one hand, the recovery of breast cancer screening programs may have been more accessible for several reasons. First of all, the diagnosis of this neoplasia is not so age-dependent, as with, for example, prostate cancer, and, possibly due to a higher concern, females may have had more access to sanitary healthcare, as previously commented upon. Breast cancer screening was also the first cancer screening program to recover in Madrid during the pandemic. Additionally, the high numbers of chest X-rays and CT scans performed for the evaluation of pulmonary COVID-19 disease may have modified the impact of incident diagnoses of lung cancer. This could explain, at least in part, better recovery following the diagnosis of both breast and lung neoplasia. Very recently, another population-based study performed in England confirmed that breast cancer diagnosis has recovered to pre-pandemic levels [25].

Changes in cancer screening were used during the COVID-19 pandemic to facilitate distancing from sanitary institutions and to cope with the blockage of the surgical waiting list and outpatient activities needed to confirm a cancer diagnosis. Stool testing was increased to counterbalance the decrease in gastrointestinal endoscopy practice [26,27], but as a screening test, it is only effective when the process is completed with colonoscopy [17]. Similarly, PSA testing is ineffective without a consequent prostate biopsy.

Apart from the surgical delay caused by the pandemic, the ability to deliver neoadjuvant therapy has also been impacted. The long-term adverse consequences of the disrupted care suffered with regard to cancer patient outcomes are obvious but difficult to quantify [28,29]. Delayed diagnoses will cause a serious burden that healthcare systems may not have the capacity to endure. Health systems should develop plans to quickly alleviate the accumulated demand. Additionally, some reports already warn that the impact of the pandemic in cancer care has already increased the rates of cancer-related deaths [30]. Similarly to what we observed, a study from the United Kingdom confirmed the worrisome figures of the evolution of colorectal cancer, which could lead to a future increase in mortality for this disease [31]. According to our data, besides the continued underdiagnosis of some of the most major malignancies since the COVID-19 pandemic started (i.e., prostate and colorectum cancer), a tendency towards an increase in diagnoses of colorectal cancer at advanced stages has been confirmed. These figures will likely change the panorama of cancer care in Madrid in subsequent years. On the other hand, breast cancer overdiagnosis in 2021 in Madrid has been accompanied by a significant reduction in advanced forms of this malignancy.

It is debatable whether the reference period length used for the study to evaluate the weight of the backlog of cancer during the pandemic is appropriate or if a longer period would be more adequate. We feel that evaluating a short period in a stable registry such as RTMAD close to the pandemic gives the lowest chance of including other confusion factors and favors the comparison on a yearly basis.

There are several other limitations to this study. Firstly, an incomplete picture of the population in Madrid may have accounted for patients who exclusively attended private clinics in Madrid during the pandemic outside the public hospitals that participate in RTMAD; however, many of these cases would have been secondarily registered after the initial diagnosis when they later attended hospitals in the public network. This limitation could be more likely in the younger age group of patients with increased use of private insurance, but this is the group with a lower cancer risk. Another restriction remains in the accuracy of the registry to evaluate the tumor extent at diagnosis, especially in cases not undergoing tumor resection and histopathological evaluation. We must also take into account that missing diagnoses more likely correspond to asymptomatic patients at an earlier stage, and this also hinders the proper evaluation of underdiagnosis of the tumor stage at presentation. Finally, the number of cases who moved their home outside of Madrid during lockdown, and were thus likely diagnosed and attended to in other communities, is unknown.

## 5. Conclusions

In Madrid, there is a large volume of undetected cancer cases related to the COVID-19 pandemic, and incidence rates have not returned yet to the reference levels. The pattern of cancer diagnosis that we observed confirms a substantial proportion of missed care for many different neoplasia. Breast and lung cancer screening programs have recovered completely, but other very prevalent malignancies such as colorectal, prostate or bladder cancer still remain severely impacted. These figures could have serious consequences in the future regarding cancer control and, ultimately, survival. In order to mitigate the long-term consequences of the COVID-19 pandemic regarding cancer, the health authorities and stakeholders will need to consider new strategies to recover the backlog of diagnoses, increase guideline-accepted cancer screening, efficiently treat the excess in diagnoses, and cope with the consequences of a likely delay in diagnosis that will be detected in the future.

## Figures and Tables

**Figure 1 cancers-15-01753-f001:**
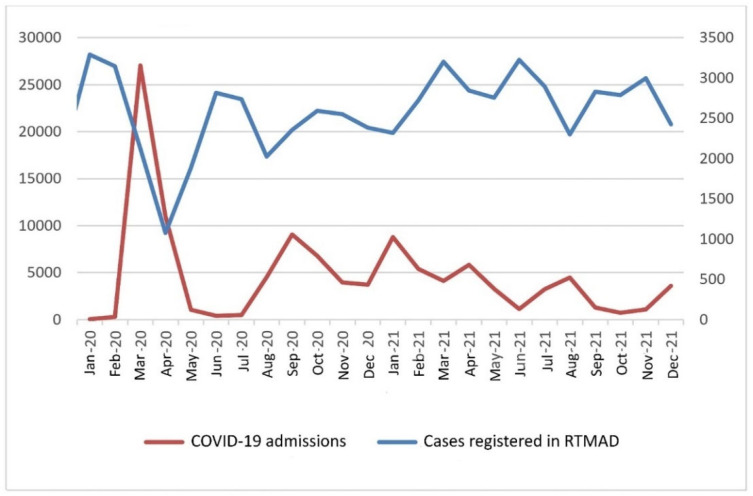
Evolution of the number of cancer diagnoses registered in RTMAD (blue line, legend right side) and the number of COVID-19 hospital admissions (red line, legend left side) during the same period (2020–2021).

**Figure 2 cancers-15-01753-f002:**
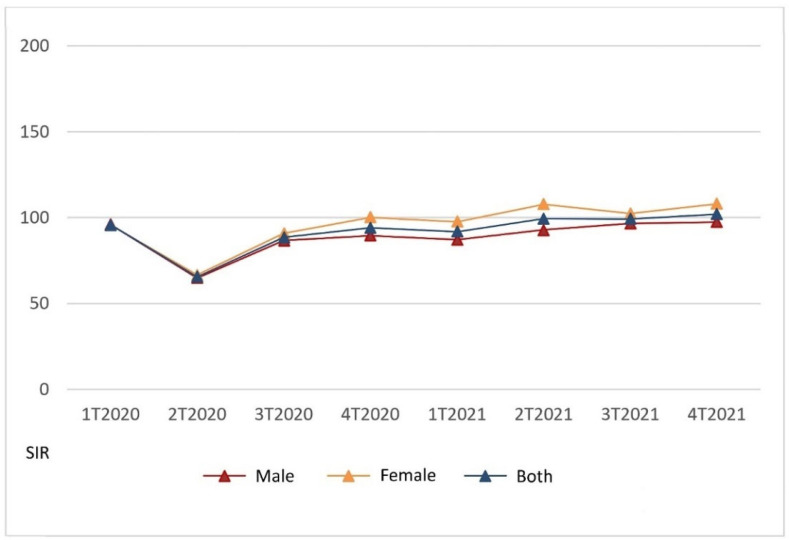
Standardized incidence ratio of cancer diagnoses registered in RTMAD per trimester for 2020–2021 (COVID-19 period) compared to 2019 (baseline period), and according to gender (male red, female yellow).

**Figure 3 cancers-15-01753-f003:**
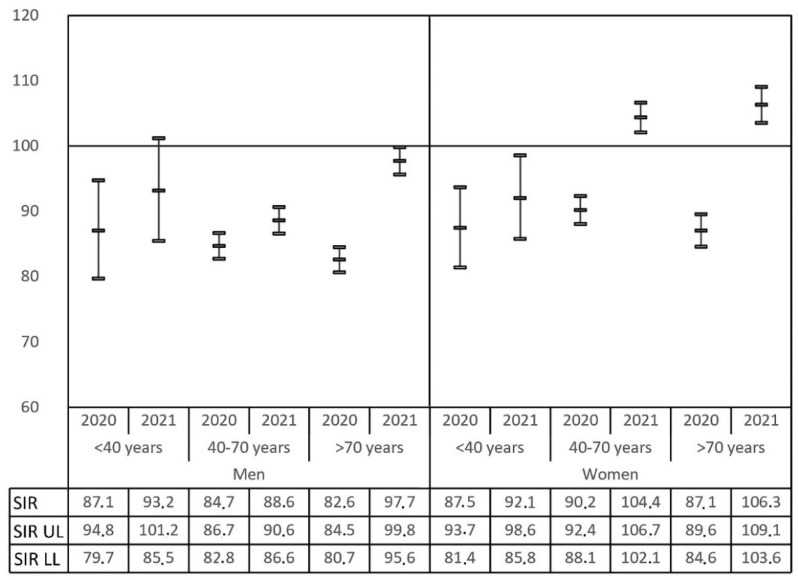
Standardized incidence ratio (SIR) of cancer diagnoses registered in RTMAD for 2020 and 2021 compared to 2019 (baseline pre-COVID-19 period) according to gender and age groups (<40 years, 40–70 years and >70 years). The SIR upper (SIR UL) and lower limits (SIR LL) of 95% confidence intervals are represented.

**Figure 4 cancers-15-01753-f004:**
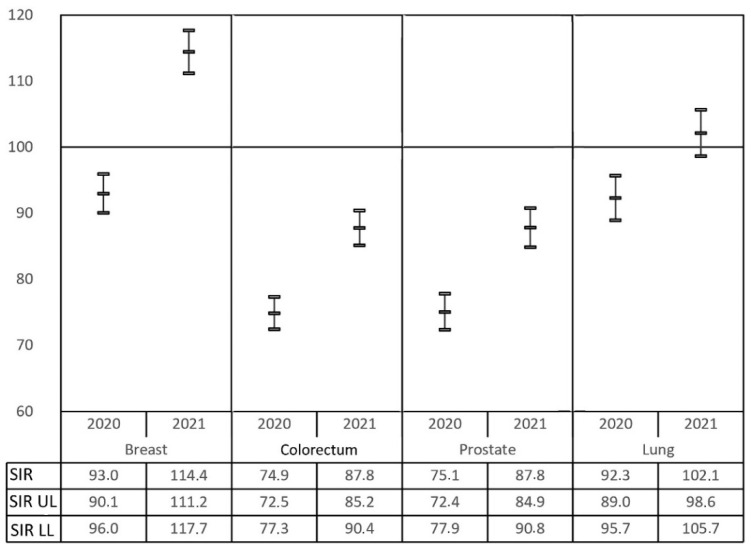
Standardized incidence ratio (SIR) of the most common cancer diagnoses registered in RTMAD for 2020 and 2021 compared to 2019 (baseline pre-COVID-19 period) according to cancer type (breast, colorectum, prostate and lung). The SIR upper (SIR UL) and lower limits (SIR LL) of 95% confidence intervals are represented.

**Figure 5 cancers-15-01753-f005:**
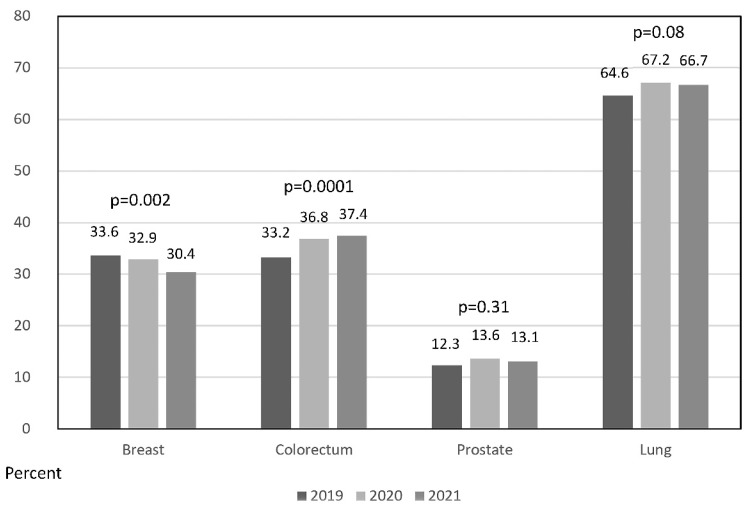
Percentage of locally advanced or metastatic cases at presentation per year during the period 2019–2021 for the most common cancer diagnoses registered in RTMAD (breast, colorectum, prostate and lung). *p*-values are expressed as Cochran–Armitage trend test.

**Table 1 cancers-15-01753-t001:** Number of cases with cancer diagnosed in RTMAD during 2019–2021 and gender and age distribution.

	2019 N (%)	2020 N (%)	2021 N (%)	*p*-Value
Total	32,976 (100)	29,699 (100)	34,279 (100)	
Male	18,300 (55.5)	15,184 (53.2)	17,654 (52.5)	<0.001 *
Female	14,676 (44.5)	13,885 (46.8)	16,625 (48.5)	
<40 years	1700 (5.2)	1937 (6.5)	2155 (6.3)	
40–70 years	16,757 (50.8)	15,177 (51.1)	17,071 (49.8)	<0.001 *
>70 years	14,519 (44)	12,585 (42.4)	15,053 (43.9)	
Age, years, mean (SD)	66.2 (14.9)	65.3 (15.5)	65.9 (15.6)	<0.001 **

SD, standard deviation; * chi-squared; ** ANOVA test.

**Table 2 cancers-15-01753-t002:** Annual evolution of number of cases, interannual monthly difference and interannual variation rate registered per month during years 2019–2021.

	Number of Cases			Interannual Difference		Interannual Variation Rate	
	2019	2020	2021	2020–2019	2021–2020	2020–2019	2021–2020
January	2894	3387	2373	493	−1014	117	70.1
February	2881	3226	2822	345	−404	112	87.5
March	2934	2177	3297	−757	1120	74.2	151
April	2854	1095	2940	−1759	1845	38.4	268
May	2906	1918	2846	−988	928	66	148
June	2845	2880	3313	35	433	101	115
July	3021	2797	2980	−224	183	92.6	107
August	2182	2069	2362	−113	293	94.8	114
September	2644	2421	2920	−223	499	91.6	121
October	3001	2667	2868	−334	201	88.9	108
November	2683	2618	3082	−65	464	97.6	118
December	2131	2444	2476	312	32	114.6	101
Total	32,976	29,699	34,279	−3278	4580	90.1	115

**Table 3 cancers-15-01753-t003:** Standardized incidence rates with 95% CIs for 2020 and 2021 compared to 2019 for different tumors according to ICD-O-3.2 topography.

Tumor Topography	N 2020	SIR 2020	95% CI 2020	N 2021	SIR 2021	95% CI 2021
Lip, oral cavity, pharynx (C00–C14)	677	92.3	85.5–99.4	728	98.5	91.5–105.8
Esophagus (C15)	210	96.7	84.1–110.2	238	108.6	95.3–122.9
Stomach (C16)	684	80.2	74.3–86.4	868	101.3	94.7–108.1
Colon, rectum (C18–C20)	3628	74.9	72.5–77.3	4288	87.8	85.2–90.4
Pancreas (C25)	780	98.1	91.4–105.1	872	108.8	101.7–116.1
Other digestive tract (C17, C21, C26)	230	90.6	79.3–102.7	249	971	85.9–110.1
Larynx (C32)	312	95.5	85.2–160.4	350	106	95.2–117.4
Lung (C34)	2866	92.3	89–95.7	3199	102.1	98.6–105.7
Thymus, heart, mediastinum (C37-C39)	100	84.6	68.8–102	103	86.7	70.7–104.2
Nasopharynx (C31, C31, C33)	52	85.2	63.2–109.9	52	84.6	63.2–109.2
Bones (C40–C41)	62	62.5	47.9–79	71	71.4	55.7–88.9
Skin (C44)	1043	83.1	78.1–88.2	1360	108.1	102.4–113.9
Retroperitoneum, peritoneum (C48–C49)	225	96.7	84.5–109.8	292	125.2	111.3–140
Breast (C50)	3859	93	90.1–96	4775	114.4	111.2–117.7
Vulva, vagina (C51–C52)	127	130	108.4–153.6	127	129.4	107.9–152.9
Cervix (C53)	240	99.1	87–112.1	273	113	100–126.9
Endometrium (C54)	599	93.7	86.4–101.4	622	96.5	89–104.2
Ovary (C56) *	326	82.5	73.8–91.7	402	100.9	91.3–111
Uterus non specified (C55, C57, C58)	41	115.3	82.7–153.3	47	131.4	96.5–171.7
Prostate (C61)	2895	75.1	72.4–77.8	3411	87.8	84.9–90.8
Testis (C62)	131	99.8	83.4–117.6	134	103.8	87–122.1
Penis (C60–C63)	47	73.5	54–96	52	81.1	60.6–104.7
Kidney, urinary tract (C64, C66–C68)	2693	82.9	79.8–86.1	2986	91.4	88.1–94.7
Renal pelvis (C65)	60	96.5	73.6–122.4	87	138.6	111–169.3
Eye (C69)	63	97.6	75–123.1	52	80.3	60–103.6
Meninges, brain, spinal cord (C70–C72)	611	89.8	82.8–97	663	97.1	89.8–104.6
Thyroid (C73)	640	76.8	71–82.99	726	87.1	80.9–93.6
Adrenal gland, other endocrine (C74, C75)	78	62	49–76.6	102	81.1	66.2–97.7
Unknown primary site (C80)	334	112.4	100.7–124.8	326	109	97.5–121.1
Lymphoid, myeloid (C42, C77)	2666	93.3	89.8–96.9	2833	98.7	95.1–102.4

SIR (95% CI): standardized incidence ratio (95% confidence interval); 95% CIs including 1 are not statistically significant; SIRs below 1 indicate significant underdiagnosis (in orange color), and over 1 indicate significant overdiagnosis (in green color). * In situ carcinoma was excluded from the analysis as it was not registered until 2020.

**Table 4 cancers-15-01753-t004:** Ranking of the most frequent tumors registered in RTMAD during 2019–2021, both for the total population and according to gender.

2019		2020		2021	
Total	N (%)	Total	N (%)	Total	N (%)
Colorectum	4051 (13.2)	Breast	3417 (12.6)	Breast	4201 (13.5)
Prostate	3692 (12.1)	Colorectum	3014 (11.2)	Colorectum	3588 (11.6)
Breast	3611 (11.8)	Prostate	2881 (10.7)	Prostate	3377 (10.9)
Lung	2994 (9.8)	Lung	2851 (10.6)	Lung	3172 (10.2)
Male	N (%)	Male	N (%)	Male	N (%)
Prostate	3692 (21.7)	Prostate	2881 (19.7)	Prostate	3377 (20.6)
Colorectum	2406 (14.1)	Lung	1936 (13.2)	Lung	2150 (13.1)
Lung	2090 (12.3)	Colorectum	1716 (11.7)	Colorectum	2035 (12.4)
Bladder	1588 (9.3)	Bladder	1237 (8.4)	Bladder	1083 (6.6)
Female	N (%)	Female	N (%)	Female	N (%)
Breast	3568 (26.3)	Breast	3381 (23.4)	Breast	4167 (28.5)
Colorectum	1645 (12.1)	Colorectum	1298 (10.5)	Colorectum	1553 (10.6)
Lung	904 (6.6)	Lung	915 (7.4)	Lung	1022 (7)
Thyroid	622 (4.6)	Endometrium	593 (4.8)	Endometrium	619 (4.2)

## Data Availability

Full data will be available upon reasonable request.

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
