# Peer review of "Impact of the COVID-19 Pandemic on Cancer Diagnosis in Madrid (Spain) Based on the RTMAD Tumor Registry (2019–2021)"

_cancers, 2023, doi:10.3390/cancers15061753_

Round 1

Reviewer 1 Report

The manuscript entitled Impact of the COVID-19 pandemic on cancer diagnosis in Madrid (Spain) based on RTMAD tumor registry (2019-2021).” intends to evaluate the weight of this backlog on community-wide scale in Madrid during the period 2020-2021, and whether a stage shift towards advanced stage has occurred.

General Critique:

From a general standpoint, the current study is lack of the standard clinical research criteria and statistical analysis to assess or validate the impaction of COVID-19. The current study is like a summarized narrative annual cancer report during COVID-19 pandemic (2019-2021). However, this is a relatively well written study, but does have some shortcomings which limit its conclusions and potential for publication in a high impact journal such as cancers.

The basic question is interesting. However, I would like to raise the following points:

1.     Reference year (2019) in the study seems to be inadequate or biased to evaluate the weight of the backlog of cancer during pandemic. Using the average of the previous 2-3 years (2017-2018) before COVID-19 outbreak might advise to investigate the trend of cancer incidence during the outbreak of COVID-19 and make the study more convincing.

2.     “Most cancer types were underdiagnosed in 2020. The tendency worsened in 2021 for colorectal and prostate cancers (87.8%), …….compared to reference pre-COVID-19 data” in the section of Abstract. The authors should be cautiously to interpreter the tendency without any statistical analysis.  

3.     Briefly to simplify the redundant description in the Result of Table 3. The table 3 could be allocated to supplemental information or only report the significant data.

4.     The figures and legends should be carefully revised in regard to abbreviated terms and add statistical analysis among study periods.    

5.     The manuscript should be carefully revised in regard to grammatical errors, and the quality of the figures, tables and the whole manuscript need to be improved.

Reviewer 2 Report

The purpose of this study is to examine the consequences of the COVID-19 pandemic on cancer diagnosis, and follow-up and on the development of neoplasia. For that, the authors used the data of 29 hospitals in Madrid using Tumor registry (RTMAD). They compared the year 2019 (as a reference) with data from the years 2020 and 2021 (COVID-19 pandemic). The study provides interesting findings, but clarifications are necessary to facilitate the understanding of the results.

Abstract: The authors wrote “Most cancer types were underdiagnosed in 2020. The tendency worsened in 2021 for colorectal and prostate cancers (87.8%), but lung cancer recovered (102.1%) and breast cancer was over-diagnosed (114.4%) compared to reference pre-COVID-19 data”. It would seem that breast cancer was over-diagnosed, but it is unclear how this may be linked to the pandemic.

Introduction: The sentence from lines 56 to 60 must be rephrased in two sentences. In the objective, the authors wrote “is to evaluate the weight of this backlog on community-wide scale during the first two years of pandemic for the different neoplasia in Madrid and how future allocation of resources for cancer health care recovery need be provided”. I am not sure that the authors examined “future allocation of resources”. The objective must be rephrased by targeting the main objective, that is, the examination of the consequences of COVID-19 on cancer diagnosis, and follow-up.

Patients and methods: First of all, the authors should explain more precisely what is the standardized Incidence Ratio (SIR) and to explain how it is used in their study. The standardized incidence ratio (SIR) is generally used when disease rates in the cohort under study are being compared to disease rates in a reference population, such as the general population of the geographic area from which the cohort was selected. Overall, and for all criteria, authors should clearly explain how all of these measures complement each other. It is also important to underline the possible confounded factors and how (if justified) they were controlled.

Results: The results of Table 1 are not described, and the authors could use inferential analysis to test the significance of the change. Moreover, the authors should add subtitles in the results section in order to clearly describe their findings and facilitate the reading. What means SIR UL and SIR LL (page 6)? The tables are not easy to understand, and the notes should add more information. 

Discussion: The first paragraph of the discussion section is not appropriate. It should rather be in the introduction. Moreover, the discussion of the findings must be improved by explaining more the supposed links between pandemic and cancer diagnosis. In the abstract, the authors highlighted the over-diagnosed breast cancer during pandemic but don’t really discuss these findings. It is not clear why this over-diagnosis.

Overall, the manuscript provides interesting information but, which are difficult to identify. The manuscript needs to be rewritten in a clearer way.

Round 2

Reviewer 1 Report

As I am not a statistician I am not able to give valuable feedback on Methods, however, the authors should be cautiously to interpret the results. In summary, the manuscript in its current form does not add any novel aspect to the current literature on Covid-19 pandemic on cancer diagnosis.

Reviewer 2 Report

I would like to thank the authors for improving their paper. Nevertheless, I still believe that the result section should have subtitles to clearly specified the results against their objectives. Moreover, there is still some grammar errors and sentence formulation that need to be corrected.

For example: Line 44 “These facts will determine how future allocation of resources for cancer health care recovery need be provided”. Replace “need” by “should”.

Line 159 “Chi-x2 test” replace by “Chi-squared”; Line 141 “The only factor that could cause an influence on the change of population structure during the years evaluated is precisely COVID-19 mortality in 2020 and, to a lesser extent, in 2021.” Replace by “The only factor that may have an influence on the change of the population structure during the years evaluated is precisely the mortality by COVID-19 in 2020 and, to a lesser extent, in 2021.”

Line 266, “Quite the contrary, statistically significant relative decrease in advanced breast malignancy has been confirmed (p=0.002).” remove “relative”.

Line 300 “The evolution of cancer diagnosis regarding according to gender has already been presented in Figure 2.” Replace “regarding according” by “by”.

Line 371 “In Madrid there is a large volume of undetected cancer cases in related to the COVID-19 pandemic and incidence rates have not returned yet to reference levels.” Remove “in”.

A proofreading of English by experts seems important to me
